# Hydrogen Sulfide Treatment Alleviates Chilling Injury in Cucumber Fruit by Regulating Antioxidant Capacity, Energy Metabolism and Proline Metabolism

**DOI:** 10.3390/foods11182749

**Published:** 2022-09-07

**Authors:** Jingda Wang, Yaqin Zhao, Zhiqian Ma, Yonghua Zheng, Peng Jin

**Affiliations:** College of Food Science and Technology, Nanjing Agricultural University, Nanjing 210095, China

**Keywords:** cucumber, hydrogen sulfide, chilling injury, antioxidant enzyme, energy, proline

## Abstract

Although low-temperature storage could maintain the quality of fruits and vegetables, it may also result in chilling injury (CI) in cold-sensitive produce, such as cucumbers. This can seriously affect their quality.” The antioxidant capacity, energy metabolism and proline metabolism of cucumbers treated with hydrogen sulfide (H_2_S) were studied in this assay. The outcomes displayed that H_2_S treatment effectively reduced CI and delayed the increase in electrolyte leakage (EL) and malondialdehyde (MDA) content. In addition, the H_2_S-treated cucumber fruit exhibited higher L* and hue angle values, as well as nutrients such as ascorbic acid (AsA). The H_2_S-treated fruit showed lower levels of reactive oxygen species (ROS) and higher antioxidant enzyme activities. Meanwhile, H_2_S treatment also increased the activities of the essential enzymes involved in energy metabolism, including cytochrome C oxidase (CCO), succinate dehydrogenase (SDH), H^+^-ATPase and Ca^2+^-ATPase, which improved the energy supply. H_2_S induced higher ornithine δ-aminotransferase (OAT) and Δ-1-pyrroline-5-carboxylate synthetase (P5CS) activities, and reduced proline dehydrogenase (PDH) activity, promoting the accumulation of proline. These results indicated that H_2_S could alleviate CI in the cucumber fruit by modulating antioxidant capacity, energy metabolism and proline metabolism, thereby extending the shelf life of postharvest cucumbers.

## 1. Introduction

Cucumber (*Cucumis sativus* L.) belongs to the Cucurbitaceae family and is a popular fruit that is rich in nutrients. Cucumber fruit is a good source of antioxidants, dietary fiber and ascorbic acid (AsA) [1]. Cold storage is one of the most important techniques applied to prevent degradation, diminish respiration and extend the preservation period of cucumber. However, cucumber is a very temperature-sensitive commodity. When the temperature is lower than 10 °C, many chilling injury (CI) symptoms, such as surface pitting, water spots, scale depressions, tissue collapse and decay, are prone to occur in cucumber [2]. CI symptoms reduce the shelf life of cucumber fruit and caused huge economic losses. For many years, many approaches have been applied to reduce CI symptoms in cucumber fruit, including salicylic acid [3], heat stimulation [4], melatonin [5] and putrescine [6]. Hydrogen sulfide (H_2_S) is universally recognized to be a poisonous gas that smells like rotten eggs. Nevertheless, there is growing evidence that only high concentrations of H_2_S are harmful to cells, while low levels of endogenous H_2_S perform a diverse range of physiological functions in horticultural products [7]. H_2_S is considered to be the third clock gas signaling molecule alongside nitric oxide (NO) and carbon monoxide (CO) [8]. Previous findings suggested that H_2_S could reduce CI in bananas [9], sweet cherries [10] and hawthorns [11]. Therefore, H_2_S is considered to have a potential role in mitigating chilling injury in cucumbers.

The intracellular energy status is thought to be a major factor in the post-harvest maturation and senescence of horticulture products [12,13]. More and more studies have shown that an enhanced energy state sustains membrane integrity [14,15]. The cell membrane is the main site of CI development [16]. Maintaining higher levels of adenosine triphosphate (ATP) and energy charge (EC) also contributes to membrane integrity [17]. Physiological responses associated with senescence, for example, enhanced membrane permeability and increased reactive oxygen species (ROS) generation, are likely to be associated with inadequate energy supply. An increasing amount of research showed that energy metabolism was linked to the occurrence of CI in plants. Under low-temperature stress, plant tissues required more energy to maintain normal life activities. However, the tissues had a decreased supply of energy due to the reduced metabolic level. The contradiction between these two aspects might contribute to CI in horticulture products [13]. Proline is an osmoregulatory substance that is widely present in plants [18]. When plants are under cold stress, proline content increases significantly to protect their tissues [19]. The accumulation of proline has a major function in cold resistance in plants [20].

However, the impacts of exogenous H_2_S treatment on antioxidant metabolism, energy metabolism and proline metabolism of cucumber fruit have not been reported. Thus, the aim of this research was to study the effect of H_2_S treatment on cucumber fruit and the mechanism associated with mitigating CI.

## 2. Materials and Methods 

### 2.1. Plant Materials and Treatments

Cucumber fruit were picked at the commercial maturity stage from a farm in Nanjing, Jiangsu, China. Then they were immediately carried back to the laboratory. Cucumbers of uniform size and no mechanical injury were picked and randomly separated into two groups (300 fruit in each group). The control group was soaked in deionized water for 10 min, while the treatment group was soaked in a 1.0 mM sodium hydrosulfide (NaHS) for 10 min. All fruit were packed in plastic bags after drying, with 5 fruit in each bag. Then, they were placed in a thermostat at a temperature of 4 °C and relative humidity of 85–90%. Sixty fruit were randomly selected every 3 days during the storage period with 3 replicates. The cucumber fruit were stored for a total of 15 days. On each sampling day, 30 fruit were placed on a rack at 20 °C for 2 days to determine the CI index. Another 30 fruit were cut into small pieces, frozen using liquid nitrogen and stored at −80 °C for the determination of other indicators.

### 2.2. CI index, Electrolyte Leakage (EL), L* Value and Hue Angle (H*) Value

The CI index of cucumber was measured based on the method of Lin et al. [21]. The fruit were ranked on the CI scale from 0 to 4, where 0—none, 1—slight, 2—moderate, 3—moderately severe and 4—severe. The CI index is presented in Equation (1):(1)$$\mathrm{CI}\;\mathrm{Index}\;=\;\frac{\sum \left(\mathrm{CI}\;\mathrm{scale}\;\times \;\mathrm{the}\;\mathrm{number}\;\mathrm{of}\;\mathrm{fruit}\;\mathrm{in}\;\mathrm{each}\;\mathrm{scale}\right)}{\left(\;4\;\times \;\mathrm{the}\;\mathrm{total}\;\mathrm{number}\;\mathrm{of}\;\mathrm{fruit}\;\right)}$$

The measurement of EL was based on the method of Ali et al. [22]. Ten small pieces of the middle part of cucumber peels were taken. The thickness of each piece was about 1 mm. The piece was placed in a test tube. The conductivity was determined with a conductivity meter after adding 25 mL of distilled water. Conductivity C_1_ was measured after standing for 30 min and then the test tube was boiled for 15 min. Conductivity C_2_ was measured after cooling. The formula is as follows (Equation (2)):(2)$$\mathrm{EL}\left(\%\right)=\frac{C_{1}-C_{0}}{C_{2}-C_{0}}\times 100\%\;$$

The L* and H* values reflect the lightness and yellow-green values of the color, respectively. Three points were picked up on the equator of each cucumber fruit and measured with a color difference meter. The value of H* was calculated using Equation (3):(3)$$H^{*}=180-\arctan\left(\frac{a^{*}}{b^{*}}\right)$$

### 2.3. Chlorophyll, Ascorbic Acid and Malondialdehyde (MDA) Content

Measurement of the chlorophyll content of cucumber peel was based on the method of Zhang et al. [3]. Two grams of cucumber peel was placed in a mortar, ground well with 80% acetone, left to stand and filtered. The absorbances at the wavelengths of 663 nm and 645 nm were measured using a spectrophotometer. The results were expressed as the content of chlorophyll contained in each gram of the sample, namely, mg g^−1^ FW. The formula for calculating the chlorophyll content was as follows (Equation (4)):(4)$$\;\mathrm{Chlorophyll}\;\mathrm{content}\;\left(\mathrm{mg}/g\right)=\left(20.29\times A_{645}+8.05\times A_{663}\right)\;/\;1000\;$$

A_645_ and A_663_ represent the absorbance values at wavelengths of 645 nm and 663 nm, respectively.

The AsA content was assayed according to Jia et al. [6]. Two grams of cucumber pulp was weighed. Then, the pulp was homogenized. The absorbance of the reactive solution at 534 nm was calculated using the o-phenanthroline method. The results were expressed as mg g^−1^ FW.

The MDA content was lightly modified based on the method of Jin et al. [23]. The absorbances at 600 nm, 532 nm and 450 nm were measured. The content of MDA was expressed as nmol g^−1^ FW. The formula for calculating MDA content is as follows (Equation (5)):(5)MDA content (nmolg)=6.45×(A532− A600)−0.56×A450 

A_532_, A_600_ and A_450_ represent the absorbance values at wavelengths of 645 nm, 663 nm and 450 nm, respectively.

### 2.4. 1,1-Diphenyl-2-Picrylhydrazyl (DPPH·) Scavenging Rate, Hydroxyl Radical (·OH) Scavenging Rate, Superoxide Anion (O_2_^·−^) Production Rate and Hydrogen Peroxide (H_2_O_2_) Content

The DPPH· scavenging rate was measured according to Al Ubeed et al. [24]. The formula was as follows (Equation (6)):(6)$$\mathrm{DPPH}\cdot \;\mathrm{scavenging}\;\mathrm{rate}\;\left(\%\right)=\left[\frac{A_{0}-{\;A}_{1}}{A_{0}}\right]\times 100$$

A_0_ and A_1_ represent the absorbance of the control and sample, respectively. 

The ·OH scavenging rate was measured according to Ma et al. [25]. The results were calculated using the following Equation (7): (7)$$\cdot \mathrm{OH}\;\mathrm{scavenging}\;\mathrm{rate}\;\left(\%\right)=\left[\frac{A_{0}-{\;A}_{1}}{A_{0}}\right]\times 100$$

The meanings of A_0_ and A_1_ are the same as in Equation (5).

O_2_^·−^ production was calculated according to Ma et al. [25]. Two grams of pulp tissue was ground in 5 mL of 50 mM phosphate buffer. The homogenate was centrifuged at 10,000× *g* for 20 min at 4 °C. The supernatant was used for the determination of O_2_^·−^ production. The absorbance value of the reaction solution at the wavelength of 530 nm was measured. A potassium nitrite (KNO_2_) standard curve was used to measure the O_2_^·−^ production. The results were expressed in µmol min^−1^ g^−1^ FW.

To measure the content of H_2_O_2_, 2.0 g of cucumber pulp was mixed with 100% acetone. The H_2_O_2_ content in the crude extract was detected according to Zhang et al. [26]. The outcome of H_2_O_2_ content was indicated in µmol g^−1^ FW.

### 2.5. Ascorbate Peroxidase (APX), Superoxide Dismutase (SOD), Catalase (CAT) and Peroxidase (POD) Activities

The activities of APX, SOD, CAT and POD were determined according to Wang et al. [27]. The amount of enzyme needed to decrease the absorbance at 290 nm by 0.01 per minute was one APX unit (U). The quantity needed to suppress NBT photoreduction by 50% per minute was used as one unit of enzyme activity (U) of SOD. The amount of enzyme that decreased the absorbance value of the reaction system at 240 nm by 0.01 per minute was considered as one CAT unit (U). One POD unit (U) was considered the quantity of enzyme needed to decrease the absorbance value of the reaction solution by 0.01 per minute at 470 nm. All anti-oxidant enzyme activities were expressed in U g^−1^ FW.

### 2.6. H^+^-ATPase, Ca^2+^-ATPase, Cytochrome C Oxidase (CCO) and Succinate Dehydrogenase (SDH) Activities

The extraction of cucumber mitochondria followed the approach of Li et al. [28]. Cucumber samples (5.0 g each) were ground with 10 mL of pre-chilled Tris-HCl and filtered through five layers of gauze. The supernatant was centrifuged to obtain the precipitate. The precipitate obtained after undergoing centrifugation again for 10 min was the mitochondria. It was placed at a low temperature for the next step of the experiment. The activities of H^+^-ATPase and Ca^2+^-ATPase were determined according to Jin et al. [29]. The measurements of SDH and CCO activities were measured according to Cheng et al. [30]. The results were expressed as U g^−1^ FW.

### 2.7. ATP, Adenosine Diphosphate (ADP) and Adenosine Monophosphate (AMP) Contents and EC

The energy level was determined according to Zhou et al. [31]. The cucumber pulp was mixed with 5 mL HClO_4_ solution and thoroughly ground. Then, the volume was increased to 6 mL with ultra-pure water. It was filtered with a 0.45 μm water filter and analyzed using high-performance liquid chromatography. The samples were qualitatively analyzed according to the retention time of the standard and quantitatively analyzed according to the peak area of the standard. The final content was expressed in µg g^−1^ FW. The calculation method for EC is as follows (Equation (8)): (8)EC(%)=(ATP +0.5×ADP)(ATP +ADP +AMP)×100

### 2.8. Proline Content and Δ-1-Pyrroline-5-Carboxylate Synthetase (P5CS), Proline Dehydrogenase (PDH) and Ornithine δ-Aminotransferase (OAT) Activities

Proline content was assayed according to Zuo et al. [18]. Cucumber samples (1.0 g each) were ground with 3% sulfosalicylic acid. Samples were extracted by shaking at 100 °C for 10 min after grinding into a homogenate. After centrifugation, the supernatant was extracted by adding acidic ninhydrin and boiled for 25 min. The reaction solution was extracted with toluene. The absorbance value of the organic phase at 520 nm was measured. The results were expressed in μg g^−1^ FW. The enzymatic activities of P5CS, PDH and OAT were measured according to Zhang et al. [26]. The enzyme solution was mixed with the reaction system via centrifugation at 12,000× *g*. The absorbance data at 340 nm were measured. The change in absorbance value by 0.001 per minute of the mixed reaction solution was considered as 1 U of enzyme activity. The results were shown as U g^−1^ FW. Reactions were initiated with 0.15 mL NAD^+^ for the determination of PDH activity. Cucumber samples were triturated with sodium phosphate buffer to measure the OAT activity. The supernatant obtained via centrifugation was mixed with the reaction system. The absorbance values at 510 nm were measured. 

### 2.9. Statistical Analysis

Experiments were performed in a completely randomized design and the results were expressed as mean ± standard deviation (SD). Origin 2022 software was used for graphing, and SAS 9.21 software was used for independent samples *t*-tests (*p* < 0.05).

## 3. Results

### 3.1. CI Index, EL, L* Value and H* Value

The CI index increased during storage. The symptoms of CI in the control group appeared earlier and were more severe than in the H_2_S-treated group (Figure 1A). EL from cucumber fruit remained elevated throughout the storage period; the increase in the H_2_S treatment group was significantly smaller than in the control group (Figure 1B).

The L* value of the cucumber peel showed an increasing trend with extended storage (Figure 1C). The L* value of the H_2_S-treated group remained lower than that of the control group except for the third day of storage. The H* value of the cucumber peel showed a decreasing trend with extended storage (Figure 1D). The H* value of the H_2_S treatment group remained higher than that of the control group for the whole storage. The L* and H* values of the H_2_S treatment group were 89.5% and 102.8% of those of the control group on day 15, respectively.

### 3.2. Chlorophyll, AsA and MDA Content 

The content of chlorophyll gradually declined during storage period and the H_2_S treatment significantly suppressed the decrease in chlorophyll content (Figure 2A). The chlorophyll content in the H_2_S treatment group was 13.8% higher than in the untreated group on day 15. As shown in Figure 2B, the AsA content in the cucumber fruit in the control group remained stable for the first 6 days of refrigeration and then decreased progressively. Except for the third and sixth days of storage, the contents of AsA in the H_2_S group remained higher than that of the control group.

The MDA content of the cucumber increased throughout storage. The H_2_S treatment significantly suppressed the accumulation of MDA content (Figure 2C). The difference in MDA content in the cucumber fruit between the two groups increased from day 6. MDA content in the control group was 28.9% higher than that in the H_2_S treatment group on day 15 of storage.

### 3.3. DPPH· Scavenging Rate, ·OH Scavenging Rate, O_2_^·−^ Production Rate and H_2_O_2_ Content

The DPPH· scavenging rate in the cucumber fruit initially increased sharply. Its peak was reached on day 6 and declined gently thereafter (Figure 3A). The ·OH radical scavenging rate was elevated continuously before day 9 and then declined gradually (Figure 3B). The DPPH· and ·OH scavenging rates in the cucumber fruit treated with H_2_S remained higher than those in the control group. 

The O_2_^·−^ production rate in the cucumber increased steadily throughout storage. The H_2_S treatment significantly slowed the increase in the O_2_^·−^ production rate during cold storage (Figure 3C). The H_2_O_2_ content of the cucumber fruit increased rapidly until day 6 and then began to decrease. Compared with the control group, the H_2_O_2_ content of the H_2_S-treated fruit was consistently lower (Figure 3D). The H_2_O_2_ content of the H_2_S-treated fruit was only 72.9% of the content in the untreated fruit on day 6.

### 3.4. APX, SOD, CAT and POD Activities

In Figure 4, the activities of APX, SOD, CAT and POD displayed an increasing trend initially, but a decrease as storage continued. APX activity peaked on day 9 and then declined rapidly. The control group was always lower than the H_2_S-treated group throughout the storage period. The APX activity of the H_2_S-treated group was 27.1% higher than the untreated group on day 9 of the cold storage (Figure 4A). The SOD activity of the untreated group was consistently lower than that of the H_2_S-treated group during storage. The SOD activity of the H_2_S-treated group was 34.8% higher than that of the untreated group on day 15 of cold storage. The H_2_S treatment significantly enhanced the CAT activity and suppressed its decrease during the storage period. The CAT activity of the H_2_S-treated group was 36.0% higher than that of the control group on day 9 (Figure 4C). The POD activity of the control group was always lower than that of the treatment group (Figure 4D). The difference between the two groups reached a maximum on day 9 of storage. At this time, the activity of the H_2_S-treated group was 23.4% higher than that of the control. 

### 3.5. H^+^-ATPase, Ca^2+^-ATPase, CCO and SDH Activities

The activity of H^+^-ATPase increased initially in the cucumber fruit. After reaching its peak, it began to decline (Figure 5A). The H^+^-ATPase activity of the H_2_S-treated group was significantly higher than that of the untreated fruit. The activity of Ca^2+^-ATPase initially increased and then started to decrease after day 6 (Figure 5B). The activity of the H_2_S treatment group was always maintained at a high level. An analogous tendency was observed in SDH and CCO. The difference was that the activities of SDH and CCO began to decline after day 9 of refrigeration. The H_2_S treatment significantly elevated the activities of all the above enzymes. The activities of H^+^-ATPase, Ca^2+^-ATPase, SDH and CCO in the H_2_S-treated group were 18.6%, 15.4%, 14.7% and 23.6% higher than those in the untreated group on day 15, respectively.

### 3.6. ATP, ADP and AMP Contents and EC

The ATP and ADP contents of the cucumber fruit gradually decreased during cold storage (Figure 6A,B). The H_2_S treatment significantly maintained the ATP and ADP contents; the H_2_S-treated group had 12.6% and 14.6% higher contents than the untreated group on day 15, respectively. The AMP content increased continuously during refrigeration and H_2_S treatment could inhibit the increase in AMP in the cucumber fruit (Figure 6C). The AMP content of the H_2_S-treated group was 11.8% lower than the untreated group on day 15. According to the changes in ATP, ADP and AMP contents in the cucumber fruit, the energy charge gradually declined over the whole storage period (Figure 6C). The H_2_S treatment significantly inhibited the decrease in the EC level.

### 3.7. Proline Content and P5CS, PDH and OAT Activities

The proline content increased remarkably during the first 12 days of refrigeration and remained at a high level thereafter (Figure 7A). The H_2_S treatment significantly promoted the accumulation of proline. During the low-temperature storage, the P5CS activity gradually increased and was higher in the H_2_S-treated group than in the untreated group (Figure 7B). The activity of PDH decreased quickly in the first 9 days of refrigeration and then decreased gently (Figure 7D). The H_2_S-treated group showed lower PDH activity compared with the untreated group. The OAT activity increased substantially until day 12, after which it started to decrease (Figure 7C). The H_2_S treatment significantly enhanced the OAT activity. The OAT activity in the treatment group was 16.2% higher than the control group on day 15 of refrigeration.

## 4. Discussion

In plants, the production of ROS and its detoxification are in homeostasis, thus maintaining low levels of ROS [32]. However, during the senescence of fruits and vegetables or when they are influenced by biotic or abiotic stresses, the homeostatic system of ROS generation and scavenging is disrupted [25]. This, in turn, leads to membrane lipid peroxidation and the disruption of biofilm structure and function. Eventually, CI symptoms are induced. Plants have enzymatic and non-enzymatic antioxidant systems that eliminate ROS [9]. The enzymatic antioxidant system in plants involves SOD, POD, APX and CAT, which effectively scavenge free radicals, such as O_2_^·−^ and H_2_O_2_, in plants [33]. SOD participates in the first level of resistance against ROS by scavenging O_2_^·−^ from cells to generate H_2_O_2_ and O_2_ [4]. H_2_O_2_ is then catalyzed by CAT and POD to generate H_2_O and O_2_ [11]. APX plays a catalytic function in the ascorbate–glutathione (AsA–GSH) cycle and their combined action can effectively scavenge ROS [33]. The non-enzymatic antioxidant system of plants includes various compounds, such as AsA GSH, phenolics, carotenoids and others [34]. Therefore, they can prevent the poisonous effects of ROS on cells. Many reports revealed that the anti-oxidant ability of fruits and vegetables was related to cold resistance. For instance, Aghdam et al. [11] discovered that H_2_S treatment could effectively increase the activities of SOD, CAT and APX in hawthorn while reducing the symptoms of hawthorn CI. Chen et al. [12] noted that 6-Benzylaminopurine treatment improved SOD, CAT and APX activities in cucumber fruit while reducing CI symptoms in cucumber. Luo et al. [9] reported that H_2_S treatment diminished oxidative damage in cucumber via increasing the activity of antioxidant enzymes to reduce ROS. In this study, the activities of antioxidant enzymes, such as SOD, CAT, APX and POD, and the scavenging rates of DPPH· and ·OH in the cucumber fruit of the H_2_S-treated group remained higher than those in the untreated group, while the contents of O_2_^·−^ and H_2_O_2_ were significantly lower than those in the untreated group. This indicated that H_2_S could effectively improve the antioxidant capacity of cucumber fruit, thereby reducing chilling injury. 

The intracellular energy supply is an essential factor in the control of fruit ripening and senescence [23]. Reduced energy levels can have negative effects on various fruits, such as physiological disorders [17]. Previous studies revealed that insufficient energy supply is related to the occurrence of CI in plants [28]. For example, the ATP content and EC level in banana declined continuously during refrigeration and the storage quality was also decreased. However, after H_2_S treatment, its ATP content and EC were improved and the symptoms of CI were relieved [35]. Cheng et al. [30] found that 1-methylcyclopropene (1-MCP) could enhance cold resistance in pear by improving enzyme activities linked to energy metabolism and keeping higher levels of ATP and EC. In this experiment, the H_2_S-treated group maintained higher ATP, ADP and EC levels and lower AMP content compared with the control group. This showed that H_2_S treatment could improve the cold resistance of cucumber by keeping a higher level of intracellular EC.

To better illustrate the involvement of H_2_S in the regulation of energy metabolism, the activities of four enzymes engaged in ATP synthesis were investigated. H^+^-ATPase, Ca^2+^-ATPase, SDH and CCO are essential enzymes of energy metabolism [31]. The trends in the activities of these enzymes reflect the function of mitochondria and their ability to synthesize energy [23]. Among them, ATPase is a type of enzyme that releases more energy by catalyzing the decomposition of ATP into ADP and phosphate ions [30]. H^+^-ATPase provides energy and power for the transport of various nutrients and ions across the membrane and maintains a relatively stable cytoplasmic pH [29]. The decrease in H^+^-ATPase activity is positively correlated with cellular energy loss. Ca^2+^-ATPase can take energy released from ATP to transfer Ca^2+^ from the cytoplasm to the mitochondria or vesicles, thus maintaining the dynamic balance of Ca^2+^ concentration and contributing to the maintenance of cell homeostasis [28]. Shifts in SDH and CCO activity have an impact on the flow of electron transport chains, which, in turn, affects energy metabolism [13]. The deactivation of energy metabolizing enzymes leads to mitochondrial dysfunction, inadequate energy production and cell death [31]. Hence, energy deficiency is related to plant CI. H_2_S contributes to the cold tolerance of banana fruit during storage and is associated with increased H^+^-ATPase, Ca^2+^-ATPase, SDH and CCO activities [35]. Liu et al. [19] reported that brassinolide could alleviate CI in bamboo shoots by enhancing enzyme activities related to energy metabolism and retaining higher levels of ATP and EC. In the current research, the activities of four key enzymes of energy metabolism were significantly higher in the H_2_S-treated group than in the untreated group, which maintained higher ATP content and EC level of the cucumber. Thus, the tolerance of the cucumber fruit to low temperature during storage was improved and the occurrence of CI of the cucumber fruit was further inhibited.

Proline is an osmoregulatory substance with hydration capacity [19]. It not only maintains the osmotic balance of cells under adversity but also protects various metabolic enzymes in cells and regulates the physiological activities of plants [9]. The main proline production pathways in plant cells are P5CS and OAT, which catalyze the synthesis of proline from glutamate and ornithine, respectively. Its main depletion pathway is PDH, which can catalyze proline degradation [20]. Many studies demonstrated that many fruit and vegetable preservation methods could increase the proline content during postharvest storage by regulating the activity of these three enzymes. It stabilizes the cellular osmotic pressure balance, which, in turn, improves the cold resistance of horticulture products. Regulation of the activity of these three enzymes induced proline accumulation, which then reduced CI in banana [9], loquat [20] and bamboo shoots [19]. Similarly, in our study, the H_2_S-treated cucumbers showed higher OAT and P5CS activities and reduced PDH activity. This enhanced the accumulation of proline, and thus, reduced the presence of CI.

## 5. Conclusions

The H_2_S treatment improved the activities of antioxidant enzymes and maintained the ROS at a low level, thereby reducing the amount of oxidative damage to cell membranes. At the same time, the H_2_S treatment could effectively improve the activity of key enzymes of energy metabolism, inhibit the decrease of ATP and ADP content and the increase of AMP content, and maintain a high EC level. The H_2_S treatment also promoted proline accumulation by regulating proline metabolism. In conclusion, H_2_S enhanced the tolerance of the cucumber fruit to CI, which may be related to the regulation of antioxidants, energy and proline metabolism.

## Figures and Tables

**Figure 1 foods-11-02749-f001:**
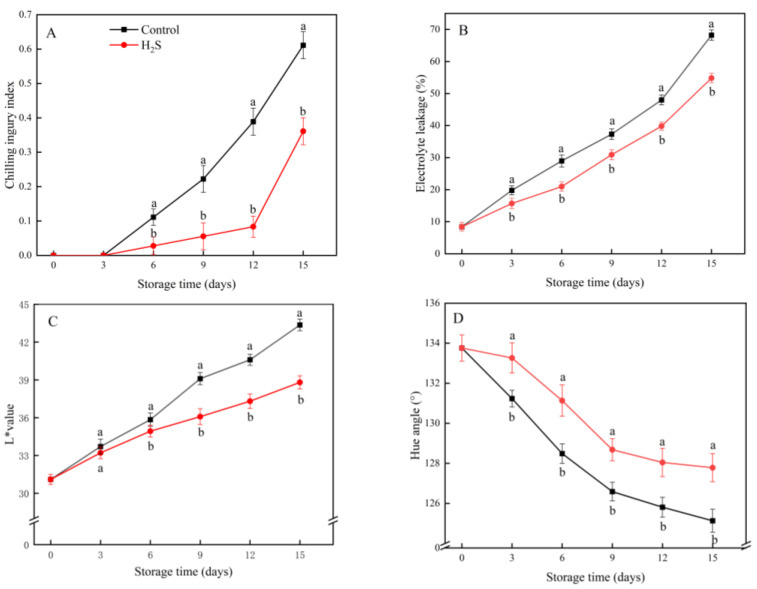
Effects of the H_2_S treatment on the chilling injury index (**A**), electrolyte leakage (**B**), L* value (**C**) and hue angle (**D**) of the cucumber stored at 4 °C over the whole storage period. Different letters represent significant differences between treatments on the same sampling day (*p* < 0.05).

**Figure 2 foods-11-02749-f002:**
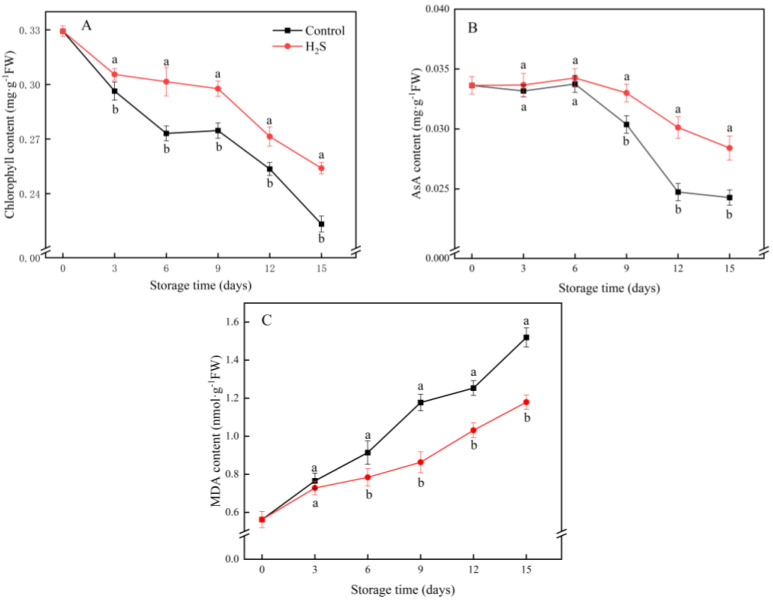
Effects of the H_2_S treatment on chlorophyll content (**A**), AsA content (**B**) and MDA content (**C**) of the cucumber fruit stored at 4 °C over the whole storage period. Different letters represent significant differences between treatments on the same sampling day (*p* < 0.05).

**Figure 3 foods-11-02749-f003:**
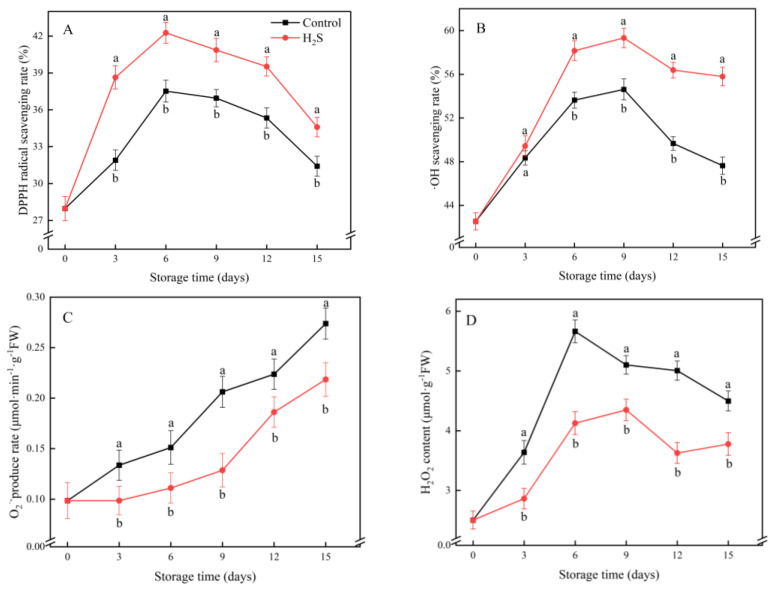
Effects of the H_2_S treatment on the DPPH· radical scavenging rate (**A**), ·OH scavenging rate (**B**), O_2_^·–^ production rate (**C**) and H_2_O_2_ content (**D**) of the cucumber fruit stored at 4 °C over the whole storage period. Different letters represent significant differences between treatments on the same sampling day (*p* < 0.05).

**Figure 4 foods-11-02749-f004:**
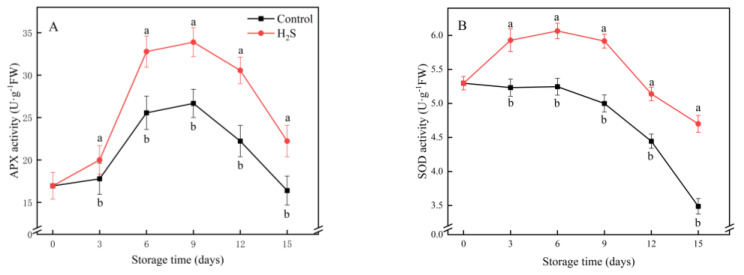
Effects of the H_2_S treatment on the activities of APX (**A**), SOD (**B**), CAT (**C**) and POD (**D**) of the cucumber stored at 4 °C over the whole storage period. Different letters represent significant differences between treatments on the same sampling day (*p* < 0.05).

**Figure 5 foods-11-02749-f005:**
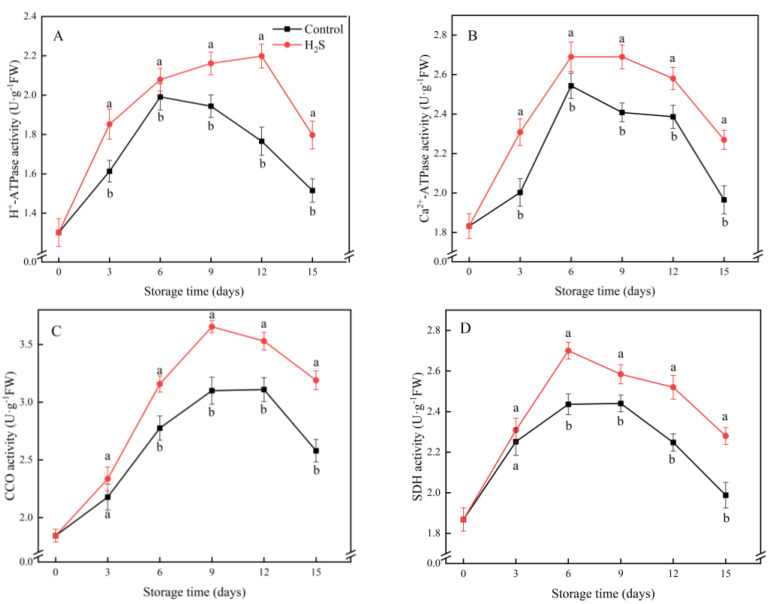
Effects of the H_2_S treatment on the activities of H^+^-ATPase (**A**), Ca^2+^-ATPase (**B**), CCO (**C**) and SDH (**D**) of the cucumber stored at 4 °C over the whole storage period. Different letters represent significant differences between treatments on the same sampling day (*p* < 0.05).

**Figure 6 foods-11-02749-f006:**
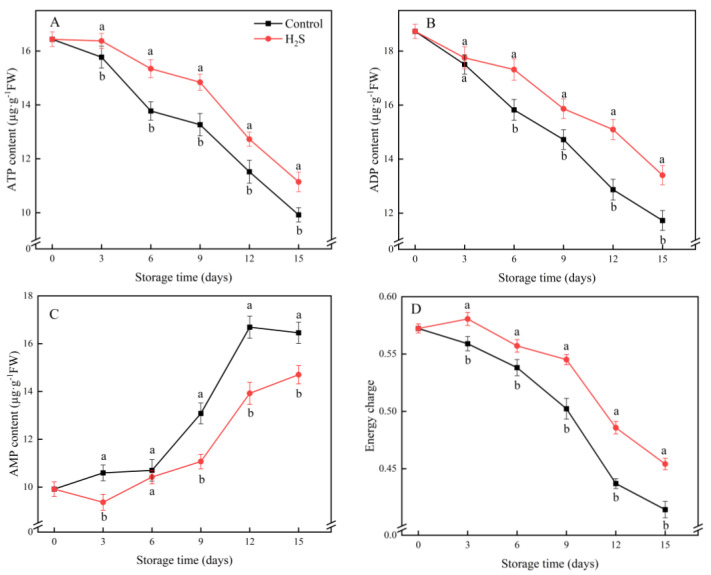
Effects of the H_2_S treatment on the activities of ATP (**A**), ADP (**B**), AMP (**C**) and energy charge (**D**) of the cucumber stored at 4 °C over the whole storage period. Different letters represent significant differences between treatments on the same sampling day (*p* < 0.05).

**Figure 7 foods-11-02749-f007:**
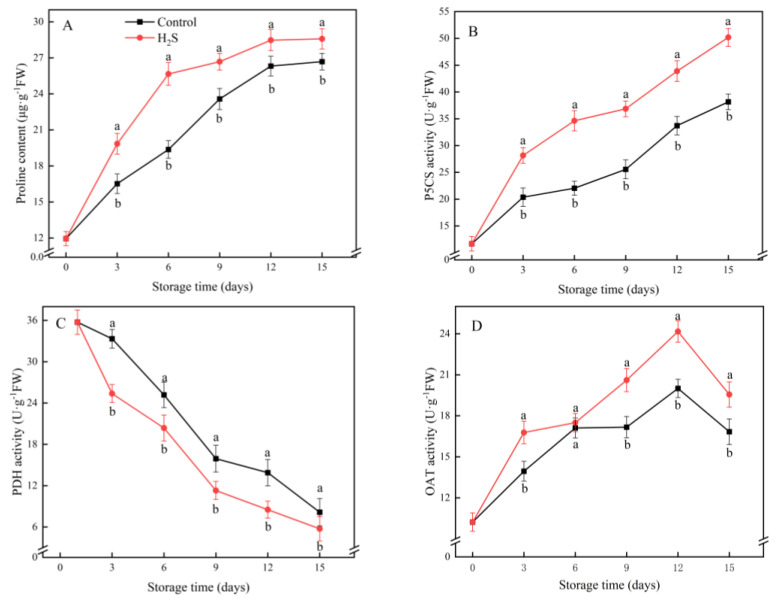
Effects of the H_2_S treatment on the proline content (**A**), P5CS activity (**B**), PDH activity (**C**) and OAT activity (**D**) of the cucumber stored at 4 °C over the whole storage period. Different letters represent significant differences between treatments on the same sampling day (*p* < 0.05).

## Data Availability

Data is contained within the article.

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
