# Peer review of "Hydrogen Sulfide Treatment Alleviates Chilling Injury in Cucumber Fruit by Regulating Antioxidant Capacity, Energy Metabolism and Proline Metabolism"

_foods, 2022, doi:10.3390/foods11182749_

Round 1

Reviewer 1 Report

In general, I found the manuscript interesting and fairly well organized and written. Figures can be improved. Here is a list of changes I suggest to apply:

L38: “sounds” should be “smells”.

L69: “it was” should be “they were”.

L76: delete “A number of”. Start the sentence with “Sixty fruit…”

L76: delete “from”.

L77-78: “A total of….”. Unclear sentence. Please rewrite.

L76-80. Unclear. A sample of 60 fruit was divided in two groups (one of 30 fruit and the other of 10 fruit)? And the other 10 fruit?

L82: the way of Liu et al? Please use a more scientific language when referring to protocols described by previous authors.

L87: Equation 1 is poorly presented. It should be rewritten following scientific standards. The summation notation (Greek letter sigma) should include a clear index of summation and the indication of the lower and upper bounds of summation. This index should be reported as subscript close to the addend/s. Please rewrite it.

L103: “was measured by the meter”. What “meter”. Please be more specific about the instruments adopted.

L106: scientists in their articles do not describe “ways”, but protocols.

L112: see my previous comment.

L125: see my previous comment.

L157: see my previous comment.

L196: “was increasing” should be “increased”.

L198: “rose” should me “increase”.

L199, L203, L213, L221, L223, etc.: it is not necessary to specify always the level of probability used (p<0.05). It is already stated in the Materials and Methods. Check this throughout the manuscript.

Figure 1: in all the panels, in the title of the x-axes put the units in brackets. I would write “(days)” instead of “(d)”. Delete also “\” before the units. When applicable, apply similar changes also in other Figures.

Figure 1B: add the units to the Y-axis. When applicable, apply similar changes also in other Figures.

Figure 1D: the title of the Y-axis is missing. When applicable, apply similar changes also in other Figures.

Line 209-210: in the caption, indicate the test used to assess the significance of the differences. When applicable, apply similar changes also in other Figures.

Line 214: 13.75%: one decimal digit is enough.

L220: “continuously” should be “progressively”.

Line 225: 28.87%: one decimal digit is enough.

Line 244: 72.86%: one decimal digit is enough.

Lines 256, 259, 261, 266, 282, 292, 296, 315, ecc. (check throughout the Results!): one decimal digit is enough.

Author Response

To Reviewer #1

In general, I found the manuscript interesting and fairly well organized and written. Figures can be improved. Here is a list of changes I suggest to apply:

Author’s response: We sincerely thank the reviewer for the favorable analysis of our original manuscript and for pointing out the shortcomings of the manuscript.

Comments:

  1. L38: “sounds” should be “smells”.

L69: “it was” should be “they were”.

L76: delete “A number of”. Start the sentence with “Sixty fruit…”

L76: delete “from”.

Author’s response: Thank you very much for pointing out some inaccuracies in the use of words in the manuscript. We have revised the manuscript according to your suggestions.

  1. L77-78: “A total of….”. Unclear sentence. Please rewrite.

Author’s response: Thank you very much for pointing this out. We rewrote the sentence as "The cucumber fruit stored for a total of 15 days."

  1. L76-80. Unclear. A sample of 60 fruit was divided in two groups (one of 30 fruit and the other of 10 fruit)? And the other 10 fruit?

Author’s response: Thank you sincerely for pointing out this noteworthy issue. On each sampling day, 60 cucumber fruits were taken from each treatment. Thirty of the fruits were placed on a rack at 20°C for 2 days to determine the CI index. Another 30 fruits were cut into small pieces, frozen in liquid nitrogen and stored at -80°C to determine other metrics. We wrote the above sentence in the "2.1. Plant materials and Treatments" position in the manuscript。

  1. L82: the way of Liu et al? Please use a more scientific language when referring to protocols described by previous authors.

Author’s response: The reviewer made a very good point here and thank for pointing out our mistakes. We use "method" instead of "way" and re-checked the whole paper.

  1. L87: Equation 1 is poorly presented. It should be rewritten following scientific standards. The summation notation (Greek letter sigma) should include a clear index of summation and the indication of the lower and upper bounds of summation. This index should be reported as subscript close to the addend/s. Please rewrite it.

Author’s response: Thank you sincerely for pointing out these noteworthy issues. We have rewritten equation 1 according to your suggestion. The new equation (1) is as follows:

  1. L103: “was measured by the meter”. What “meter”. Please be more specific about the instruments adopted.

Author’s response: Sincerely thank you for pointing out this error. There was a mistake in our manuscript, we used "spectrophotometer" instead of "meter".

  1. L106: scientists in their articles do not describe “ways”, but protocols.

L112: see my previous comment.

L125: see my previous comment.

L157: see my previous comment.

Author’s response: Thank you for pointing this out. We use "method" instead of "way" and re-checked the whole paper.

  1. L196: “was increasing” should be “increased”.

L198: “rose” should me “increase”

L220: “continuously” should be “progressively”.

Author’s response: Thank you very much for pointing out some inaccuracies in the use of words in the manuscript. We have revised the manuscript according to your request.

  1. L199, L203, L213, L221, L223, etc.: it is not necessary to specify always the level of probability used (p<0.05). It is already stated in the Materials and Methods. Check this throughout the manuscript.

Author’s response: Thank you for pointing out this noteworthy problem. We corrected this in the manuscript. We deleted duplicate (P < 0.05) and re-checked the whole paper.

  1. Figure 1: in all the panels, in the title of the x-axes put the units in brackets. I would write “(days)” instead of “(d)”. Delete also “\” before the units. When applicable, apply similar changes also in other Figures.

Author’s response: Thank you for pointing out this noteworthy problem. We changed the x-axis unit in the picture according to your request, and we checked and modified the rest of the pictures again.

  1. Figure 1B: add the units to the Y-axis. When applicable, apply similar changes also in other Figures.

Author’s response: Sincerely thank you for pointing out this error. We added units on the Y-axis of Figure 1B according to your requirements and re-checked other Figures.

  1. Figure 1D: the title of the Y-axis is missing. When applicable, apply similar changes also in other Figures.

Author’s response: Thank you very much for pointing out our mistakes. We added the Y-axis title of Figure 1D and re-checked the whole paper.

  1. Line 209-210: in the caption, indicate the test used to assess the significance of the differences. When applicable, apply similar changes also in other Figures.

Author’s response: Thank you for pointing out this noteworthy problem. In "2.9. Statistical analysis", we explained the experimental methods for assessing the significance of differences. We used independent samples T-test (P < 0.05) to assess the significance of the difference.

  1. Line 214: 13.75%: one decimal digit is enough.

Line 225: 28.87%: one decimal digit is enough.

Line 244: 72.86%: one decimal digit is enough.

Lines 256, 259, 261, 266, 282, 292, 296, 315, ecc. (check throughout the Results!): one decimal digit is enough.

Author’s response: Thank you very much for pointing out the problem of decimal numbers. We changed the digit according to your request and re-checked the whole paper.

Reviewer 2 Report

The manuscript presents new and important findings obtained in course of a comprehensive study. These findings are valuable. However, the presentation quality needs improvement in order to make the material more complete. Below are some recommendations for improving the manuscript quality and clarity.

1.         The manuscript demands a rigorous English language edition; it contains many inaccuracies in English grammar and word use.

2.         The sentence in l. 28-29 about the cucumber nutritive value seems exaggerated. Even if it refers to another publication, the latter paper does not justify any of these claims. The nutritional value claims are not essential for the present paper, but if the authors want to keep some of them, they need to provide real proofs, or omit the claims.

3.         Chlorophyll is mentioned in the abstract as a nutrient (l. 15). Is that true?

4.         Color measurement: Hue angle is mentioned in the abstract (l. 14) while M&M present L* and b* parameters without mentioning a Hue angle. Anyhow, the measurement methodology has not been presented properly (instrument, color space, measurement method/parameters, etc.). L* shows lightness, not brightness (l. 96). Why a* parameter was not used? The parameter a* is known to reflect the green color (in its negative range).

5.         The hypothetic explanation of CI development mechanism through imbalance of energy (l. 53-56) may be valid, but it differs of widely accepted theory associating a primary CI event with membrane damage. It might be recommended to present the hypothesis in a more cautious form, instead of "might be the reason…" as "might contribute to…" It would be good to present references (if available) supporting the authors' CI hypothesis.

6.         What does it mean "concentration of cytosol" (l. 60)? Please support this statement by reference/s.

7.         Some spectrophotometric methods described in M&M are not reproducible without giving equations to calculate the content of the substance analyzed. For example, the methods for chlorophyll (l. 102-105) and MDA (l. 112-114) need equations.

8.         Some methods mentioned need at least a brief explanation of their methodology, e.g. the hydrogen peroxide determination method (l. 129). What this method is based on?

9.         The reference to the DPPH method is "Al Ubeed et al." and not "H.M.S. et al." (l. 116).

10.     Please specify what does it mean "at low temperature" at l. line 145. What temperature?

11.     Instead of "the way of…" with the reference, use "the method of" or "the procedure of".

12.     All abbreviations, and in particular the enzyme names, must be explained with the full names when used in the manuscript for the first time.

13.     Axis Y titles are missing in Fig.4 B&D.

14.     In some cases, e.g. with soluble protein content, the values first increase to a maximum, and then decline? Could the authors provide an interpretation of such pattern, and in particular of the higher accumulation of the soluble protein in H2S-treated samples, and of its effect on the product's quality?

15.     What else non-enzymatic antioxidants, besides ascorbic acid, can be mentioned, even if they were not measured in this study (l. 334-335)?

16. I would recommend removing the first sentence in the "Conclusions" section (l. 401-403) because it is not a finding of the research but just a suggestion, and as a suggestion, it is presented already in l. 409-410. 

Author Response

To Reviewer #2

The manuscript presents new and important findings obtained in course of a comprehensive study. These findings are valuable. However, the presentation quality needs improvement in order to make the material more complete. Below are some recommendations for improving the manuscript quality and clarity.

Author’s response: We are honored by the reviewers' recognition of our research and thank you for your insightful and useful recommendations. We have revised them according to your suggestions.

Comments:

  1. The manuscript demands a rigorous English language edition; it contains many inaccuracies in English grammar and word use.

Author’s response: Thank you. We're sorry to have made such a mistake. We have corrected and rechecked the manuscript for inaccuracies in English grammar and word use.

  1. The sentence in l. 28-29 about the cucumber nutritive value seems exaggerated. Even if it refers to another publication, the latter paper does not justify any of these claims. The nutritional value claims are not essential for the present paper, but if the authors want to keep some of them, they need to provide real proofs, or omit the claims.

Author’s response: Sincerely thanks for your helpful suggestion. After consideration, we finally deleted the claim on the nutritional value of cucumbers according to your request.

  1. Chlorophyll is mentioned in the abstract as a nutrient (l. 15). Is that true?

Author’s response: Thank you very much for pointing out our mistakes. We deleted chlorophyll from this sentence in the abstract.

  1. Color measurement: Hue angle is mentioned in the abstract (l. 14) while M&M present L* and b* parameters without mentioning a Hue angle. Anyhow, the measurement methodology has not been presented properly (instrument, color space, measurement method/parameters, etc.). L* shows lightness, not brightness (l. 96). Why a* parameter was not used? The parameter a* is known to reflect the green color (in its negative range).

Author’s response: Thank you sincerely for pointing this out. This was our mistake in writing the manuscript. We deleted Hue angle from this sentence of the abstract. We measured the a* in the experiment, but it was not presented in the manuscript because the effect was not obvious.

  1. The hypothetic explanation of CI development mechanism through imbalance of energy (l. 53-56) may be valid, but it differs of widely accepted theory associating a primary CI event with membrane damage. It might be recommended to present the hypothesis in a more cautious form, instead of "might be the reason…" as "might contribute to…" It would be good to present references (if available) supporting the authors' CI hypothesis.

Author’s response: Thank you for pointing out this noteworthy problem. We revised the manuscript and added references according to your requirements.

  1. What does it mean "concentration of cytosol" (l. 60)? Please support this statement by reference/s.

Author’s response: Thanks, we are sorry to make such a mistake. There are wrong words in our manuscript. We have deleted this sentence from the manuscript.

  1. Some spectrophotometric methods described in M&M are not reproducible without giving equations to calculate the content of the substance analyzed. For example, the methods for chlorophyll (l. 102-105) and MDA (l. 112-114) need equations.

Author’s response: Thank you for pointing out this noteworthy problem. We added equations to the methods for determining chlorophyll and MDA in the manuscript and re-checked the whole manuscript.

  1. Some methods mentioned need at least a brief explanation of their methodology, e. g. the hydrogen peroxide determination method (l. 129). What this method is based on?

Author’s response: Thank you sincerely for pointing this out. We briefly described some mature experimental methods. We added some detailed descriptions of the experimental methods according to your suggestions.

  1. The reference to the DPPH method is "Al Ubeed et al." and not "H.M.S. et al." (l. 116).

Author’s response: Thanks, we are sorry to make such a mistake. We revised the manuscript according to your requirement.

  1. Please specify what does it mean "at low temperature" at l. line 145. What temperature?

Author’s response: Thank you sincerely for pointing this out. This "at low temperature" means 4 ℃. We made modifications in the manuscript.

  1. Instead of "the way of…" with the reference, use "the method of" or "the procedure of".

Author’s response: Sincerely thanks for your helpful suggestion. We revised and rechecked the whole manuscript according to your requirements.

  1. All abbreviations, and in particular the enzyme names, must be explained with the full names when used in the manuscript for the first time.

Author’s response: Thank you very much for pointing out our mistakes. We have revised and rechecked the entire manuscript at your request.

  1. Axis Y titles are missing in Fig.4 B&D.

Author’s response: Thank you very much for pointing out our mistakes. We added the Y-axis title of Fig.4 B&D. and re-checked the whole paper.

  1. In some cases, e. g. with soluble protein content, the values first increase to a maximum, and then decline? Could the authors provide an interpretation of such pattern, and in particular of the higher accumulation of the soluble protein in H2S-treated samples, and of its effect on the product's quality?

Author’s response: Thank you sincerely for pointing this out. Soluble proteins are important osmotic adjustment substances and nutrients. Their increase and accumulation can improve the water retention capacity of cells, protect the cells' living substances and biofilms, and are often used as indicators for cold resistance (Chen et al., 2012). The increase of soluble protein in the early stage may be due to the fact that the fruit enters the low temperature environment to translate a large amount of protein for self-protection. At the later stage of storage, with the large consumption of energy, the soluble protein content also began to decrease. The higher accumulation of soluble protein in the samples treated with H2S was mainly due to the enhancement of cold resistance of cucumber fruits and thus the reduction of cold injury.

Chen, J., Huang, S., LU, W., Chen, S., Fu, Y., & Yue, X. (2012). Study on chitosan diguanidinylated coating preservation of longan and its antifungal activities. Science and Technology of Food Industry, 33(18), 328–331

  1. What else non-enzymatic antioxidants, besides ascorbic acid, can be mentioned, even if they were not measured in this study (l. 334-335)?

Author’s response: Thank you sincerely for pointing this out. In addition to ascorbic acid (ASA), we supplemented the manuscript with non-enzymatic antioxidants such as glutathione (GSH) and carotenoid.

  1. I would recommend removing the first sentence in the "Conclusions" section (l. 401-403) because it is not a finding of the research but just a suggestion, and as a suggestion, it is presented already in l. 409-410. 

Author’s response: Sincerely thanks for your helpful suggestion. We deleted the first sentence in the "conclusion" section according to your recommendation.